# Influence of Medial Osteotomy Height and Hinge Position in Open Wedge High Tibial Osteotomy: A Simulation Study

**DOI:** 10.3390/diagnostics12102546

**Published:** 2022-10-20

**Authors:** Grégoire Thürig, Alexander Korthaus, Jannik Frings, Markus Thomas Berninger, Karl-Heinz Frosch, Matthias Krause

**Affiliations:** 1Department of Trauma and Orthopaedic Surgery, University Medical Center Hamburg—Eppendorf, 20251 Hamburg, Germany; 2Department of Orthopaedic Surgery and Traumatology, Fribourg Cantonal Hospital, University of Fribourg, 1700 Fribourg, Switzerland; 3Department of Trauma Surgery, Orthopaedics and Sports Traumatology, BG Hospital Hamburg, 21033 Hamburg, Germany

**Keywords:** HTO, medial open wedge tibial osteotomy, tibial osteotomy, loss of correction, pitfall

## Abstract

(1) Background: In treating medial unicompartmental gonarthrosis, medial open wedge high tibial osteotomy (mOWHTO) reduces pain and is intended to delay a possible indication for joint replacement by relieving the affected compartment. This study aimed to investigate the influence of the osteotomy height with different hinge points in HTO in genu varum on the leg axis. (2) Methods: Fifty-five patients with varus lower leg alignment obtained full-weight bearing long-leg radiographs were analyzed. Different simulations were performed: Osteotomy height was selected at 3 and 4 cm distal to the tibial articular surface, and the hinge points were selected at 0.5 cm, 1 cm, and 1.5 cm medial to the fibular head, respectively. The target of each correction was 55% of the tibial plateau measured from the medial. Then, the width of the opening wedge was measured. Intraobserver and interobserver reliability were calculated. (3) Results: Statistically significant differences in wedge width were seen at an osteotomy height of 3 cm below the tibial plateau when the distance of the hinge from the fibular head was 0.5 cm to 1.5 cm (3 cm and 0.5 cm: 8.9 +/− 3.88 vs. 3 cm and 1.5 cm: 11.6 +/− 4.39 *p* = 0.012). Statistically significant differences were also found concerning the wedge width between the distances 0.5 to 1.5 cm from the fibular head at the osteotomy height of 4 cm below the tibial plateau. (4 cm and 0.5 cm: 9.0 +/− 3.76 vs. 4 cm and 1.5 cm: 11.4 +/− 4.27 *p* = 0.026). (4) Conclusion: A change of the lateral hinge position of 1 cm results in a change in wedge width of approximately 2 mm. If hinge positions are chosen differently in preoperative planning and intraoperatively, the result can lead to over- or under-correction.

## 1. Introduction

For treating medial unicompartmental gonarthrosis, medial opening wedge high tibial osteotomy (mOWHTO) is an established procedure [1,2]. In addition to reducing pain, mOWHTO initially offers the option of foregoing knee arthroplasty, especially in young patients. In recent years, as well as good long-term results, a return to sports rate of over 80% has been achieved [3]. However, according to Hui et al., approximately every second mOWHTO is converted to a total knee arthroplasty after 15 years [4].

In addition to already described patient-specific risk factors, such as overweight and age, the accuracy of the correction is essential since the long-term results of mOWHTO depend mainly on postoperative alignment [5,6,7]. Therefore, in addition to the correct indication, including patient-specific risk factors, and the intraoperative procedure, preoperative planning plays a significant role in the success of the HTO. Regarding this, it has been described that the previously planned correction degree was not observed in the postoperative control in up to 70% [8,9].

Furthermore, to changes in the leg axis, mOWHTO may also result in changes in the sagittal plane of the knee joint. A biplanar descending mOWHTO can be performed to reduce the risk of an accidental change in the tibial slope [8]. Similarly, several studies have shown that the position of the hinge point influences the posterior tibial slope [10,11,12]. Moreover, the hinge fracture’s influence on correction loss and pseudarthrosis is well known [13,14] So far, no studies have investigated the impact of different osteotomy heights and different hinge positions on the wedge size. However, bone biology suggests that the smallest possible wedge size and the largest cancellous bone surface area promote rapid bone healing [15]. In addition, changes in wedge width may result in over- or under-correction compared to preoperative planning. Therefore, it is crucial for the surgeon to perform precise preoperative planning and adhere to it during the operation. A change in the hinge position can lead to an unwanted deviation from the planned leg axis. Thus, this study aimed to determine the effects of different hinge points and the osteotomy height on the resulting wedge width in mOWHTO in genu varum.

## 2. Materials and Methods

### 2.1. Radiographic Assessment

Digital radiographs were obtained from 55 patients with varus leg alignment (≥3°). According to Paley et al., full-weight-bearing long-leg radiographs of the whole lower extremity were obtained [16]. Patients stood in a weight-bearing standing position: upright with lower extremities at shoulder width, knees in full extension, and second metatarsals pointing directly toward the X-ray tube. The digitally processed antero-posterior radiographs were obtained in one exposure with a tube-to-plate distance of 2 m. Radiographs were only accepted for further analysis if the requirements were fully implemented: central patellar tracking, coverage of the fibular head by the tibia (2/3; 1/3), the position of the ankle, neutral rotation of the leg with correct visualization and projection of the trochanter minor.

Deformity planning for corrective osteotomies for the coronal and sagittal plane included a reference ball (Ø 25.4 mm) according to a three-point method (three points define a circle) for calibration. The software calculated the different angles of the lower leg based on the various reference points (hip-center, apex of the greater trochanter, medial/lateral condyle and epicondyles of femur and tibia, respectively, and medial and lateral limits of the talus).

Six groups were compared. First, mOWHTO with two different medial osteotomy heights were planned at 3 cm (Figure 1) and 4 cm (Figure 2) distal to the medial tibial plateau. Both osteotomy heights allow proper plate positioning. Secondly, an osteotomy line connecting to the tip of the fibular head was drawn. In the literature, different hinge points are recommended [17]. The results were also different concerning the correction accuracy and the risk of hinge fractures. The study of Nakamura [17], however, could show that positioning the hinge point in a “safe zone” reduces the risk of hinge fracture. For this reason, the different hinge points in this simulation study were all selected in the “Safe Zone” area. Therefore, different hinge points at 5 mm (red cross marked with an “A” in Figure 1 and Figure 2), 10 mm (yellow cross marked with a “B” in Figure 1 and Figure 2), and 15 mm (cyan cross marked with a “C” in Figure 1 and Figure 2) from the lateral wall of the tibial plateau was set which are oriented in the “safe zone” and described by the literature [17]. In each case, the planning was executed to the pre-defined point at 55% of the tibial plateau measured from the medial, which corresponds to the tip of the lateral intercondylar spine. The resulting medial osteotomy wedge opening was then measured. A difference of ≥2 mm between the measured wedge width was considered relevant from a clinical point of view. The length of the tibia and femur was determined in the whole leg radiograph and by the anatomic axis of the bones.

Two blinded observers (observer 1: experienced with digital osteotomy planning, observer 2: unexperienced) measured each radiograph with the digital planning software program (TraumaCAD© Inc. Brainlab). This program uses landmark-based approaches for alignment and deformity analysis. All preoperative radiographs were planned to a mechanical femorotibial angle (mFTA) at 55% of the tibial plateau measured from the medial. The measurements were repeated after three weeks for intra-rater reproducibility.

### 2.2. Statistical Analysis

A statistical power analysis was performed for sample size estimation. The *effect size* in this study was 0.2, considered to be small using Cohen’s criteria [18]. With a significance level *alpha* = 0.05 and *power* = 0.80, the projected sample size needed with this effect size is *n* = 55 for this simplest within-group comparison. The retrospective analysis was performed on anonymized data that had been previously collected independently of the study as part of the clinical standard. Thus, no consultation with the ethics committee is necessary.

Patients’ names and identifying features were blinded to minimize recall bias. To assess the intraobserver reproducibility and interobserver reliability, the intraclass correlation coefficient (ICCs) was calculated, including 95% confidence intervals (CIs) based on a mean-rating (k = 2), absolute agreement, 2-way mixed-effects model. The ICC was graded as ICC < 0 for *poor*; 0 to 0.2 for *slight*; 0.21 to 0.4 for *fair*; 0.41 to 0.6 for *moderate*; 0.61 to 0.8 for *substantial*; and >0.80 for (*almost*) *perfect* agreement [19].

A one-way ANOVA was performed to compare the effect of osteotomy heights and hinge points (3 cm-A, 3 cm-B, 3 cm-C, 4 cm-A, 4 cm-B, and 4 cm-C) on the wedge width. Tukey’s HSD test for multiple comparisons was then performed. A Pearson correlation coefficient was calculated to assess the linear relationship between wedge width, femoral length, and tibial length.

Statistical analysis has been performed using IBM SPSS Version 26.0 (SPSS Inc., Chicago, IL, USA).

## 3. Results

The mean age was 52.84 (24–84) years. Thirty-one right and twenty-four left full-weight bearing long-leg radiographs were analyzed. The mean mFTA was 5.95° (3°–17°) of varus. Other descriptive measurements are listed in Table 1 and Table 2. Intraobserver reproducibility and interobserver reliability were (*almost*) *perfect* for all measurements (Table 3).

Tukey’s HSD Test for multiple comparisons found that the mean value, for all measurements, of the wedge width was significantly different between the group 3 cm-A and 3 cm-C (*p* = 0.012, 95% C.I. = −3.562, 0.911), between 4 cm-A and 4 cm-C (*p* = 0.026), 95% C.I. = −4.6513, −0.178), 3 cm-A and 4 cm-C (*p* = 0.025, 95% C.I. = −4.6604, −0.187), and between 4 cm-A and 3 cm-C (*p* = 0.012, 95% C.I. = −4.838,−0.365). There was no statistically significant difference in all other compared groups (Table 4)

There was no correlation between the different wedge widths and femoral and tibial lengths.

## 4. Discussion

The main result of this simulated study shows that a shift of the hinge point by 10 mm leads to a significant change in the wedge size, which should be acknowledged in the operating theatre. Hence, reproducing the preoperative planning during surgery seems crucial to achieving a high osteotomy accuracy.

The goals of mOWHTO are to reduce pain, improve the patient’s activity level, and delay the progression of osteoarthritis to decrease the need for knee replacement arthroplasty [20]. Achieving proper lower limb alignment in mOWHTO is essential to good long-term clinical outcomes [1,21,22]. Even if the bony correction is performed as accurately as preoperatively planned, correction errors of the mechanical axis still occur [23,24,25]. Nowadays, digital programs are the gold standard in preoperative preparation for planning HTO. The planning software has shown high interrater reproducibility [26,27,28] applying the Miniaci method [29]. Also, the Dugdale method [30] is a simple planning technique that also reveals high reliability and reproducibility [27,28,31], but is not recommended due to incorrect geometry leading to underestimation [32].

The literature recommends hinge points of 0 mm, 5 mm, 10 mm, and 15 mm from the lateral cortical margin and at a variable level of the proximal tibiofibular articulation [33,34,35,36,37,38,39,40]. The authors justify their choice of hinge position because mitigation of hinge fractures is associated. Hinge fractures can cause a delay in adhesion and a loss of correction. Nakamura et al. performed a retrospective analysis with a cohort of 111 patients in order to evaluate the hinge position, which reduces the risk for hinge fractures most [17]. A “safe zone” was identified to be at the level of the proximal tibiofibular joint lateral to its margin. The hinge points selected in this study were all located in this “safe zone,” reflecting a realistic relationship to practice. Despite the fact that all the chosen hinge points are located in the “safe zone”, our data show that if the surgeon changes the position of the hinge compared to his planning because of fear regarding hinge fracture, there may also be a risk of loss of correction.

Regarding the osteotomy height, no absolute recommendation exists. Neither are there studies in which different osteotomy heights with different joint points have been analyzed [28]. This simulated study showed that the position of the lateral hinge seems crucial. Elson et al. found that when observers were free to choose the hinge point, this led to different results and was thus probably considered too vague. For this reason, they predetermined the hinge point in the second run. Mihalko et al. [41] showed that the rule of thumb of 1 mm wedge width per 1° correction does not match all knees. In this study, a change in hinge position of 1 cm can result in a change in wedge width of approximately 2 mm. Even if this is only an approximate rule of thumb, the 2 mm change in wedge width due to the different hinge positions could lead to a correction error.

In the case of a correction utilizing a wedge of the same size, there will be an under-correction for a wide compared to a narrow tibia. Hernigou [42] developed a trigonometric chart, which calculates the wedge width depending on desired correction angle and the sawing depth. In the analysis of his table, there is a corresponding difference of mostly 2 mm with a 10 mm difference in osteotomy depth and the same correction angle, which is consistent with the results of the present study.

In the discussion about problems with mOWHTO, one of the main factors is the loss of correction. Up to 2° loss of correction has been described. Kumagai et al. [7] reported that one possible cause of immediate postoperative loss of correction is the influence of the soft tissues since their configuration is different under anesthesia than in the awake state. However, they also admit the possibility of bony correction losses. Schröter et al. [26] reported that the difference between the planned correction angle and postoperative corrected angle in mOWHTO was 0.8° ± 2.0°. This under-correction was attributed to a failure to account for saw blade thickness. However, this can also be due to an intraoperative hinge position deviating from the planning, as in the following case (Figure 3a–d).

Although correction losses due to changes in osteotomy gap width are significantly lower since the introduction of rigid locking plate systems [33], changes in osteotomy gap width still have an impact. Other bony factors, such as the hinge and its position, continue to have an influence. In this context, fracture of the hinge is also a complication of mOWHTO. According to previous studies, a correlation with the wedge size could be found [40,43]. The presented data closes the loop since a larger wedge must be selected when changing the hinge position medially in order to maintain the planned axial correction. However, a larger wedge increases the fracture risk and correction loss [40].

Van de Pol et al. [44] could show at least a 1-degree deviation by the wedge width postoperatively compared to the previous planning. They attributed this to the possible deviation of the osteotomy’s width, depth, and direction from the ideal situation. They suggested that changing the positioning of the hinge, placing the wedge slightly lower, or performing an incomplete osteotomy may lead to overcorrection. The results of this study only partially support this assumption. Intraoperative changes of the preoperatively selected hinge point above 1 cm may lead to over- or under-correction, respectively. Thus, it can be hypothesized that the hinge point should be set at 5 mm medial to the lateral corticalis of the tibial plateau to avoid under- or overcorrection. However, this hypothesis needs to be tested by further studies.

Furthermore, there was no relevant difference regarding the medial sawing point. This, therefore, allows for some intraoperative flexibility. Nevertheless, the results support the need for digital preoperative planning in order to increase the success rate and reduce unintended outcomes of this procedure. Although other sources of error, such as malrotation or a flexed whole-leg radiograph, could be a possible cause of a discrepancy between the preoperative planning and the postoperative result.

The limitations of this study are that it is a simulation without a comparison group and that the measurements are based on a simulation and not on postoperative measurements. The goal was to present a possible explanation for a loss of correction, which may occur when mOWHTO is performed. Other limitations of this study are that only two observers took the measurements with different experiences in digital planning. Nevertheless, specified information in the planning could show high reproducibility and reliability.

## 5. Conclusions

The positioning of the hinge influences the wedge width. If the hinge positions are chosen differently in the preoperative planning and intraoperatively, the result can lead to over- or under-correction. A change in hinge position of 1 cm can result in a change in wedge width of approximately 2 mm. This study indicates that planning can already prevent potential errors, which is mandatory not only for postoperative success but also for medico-legal reasons.

## Figures and Tables

**Figure 1 diagnostics-12-02546-f001:**
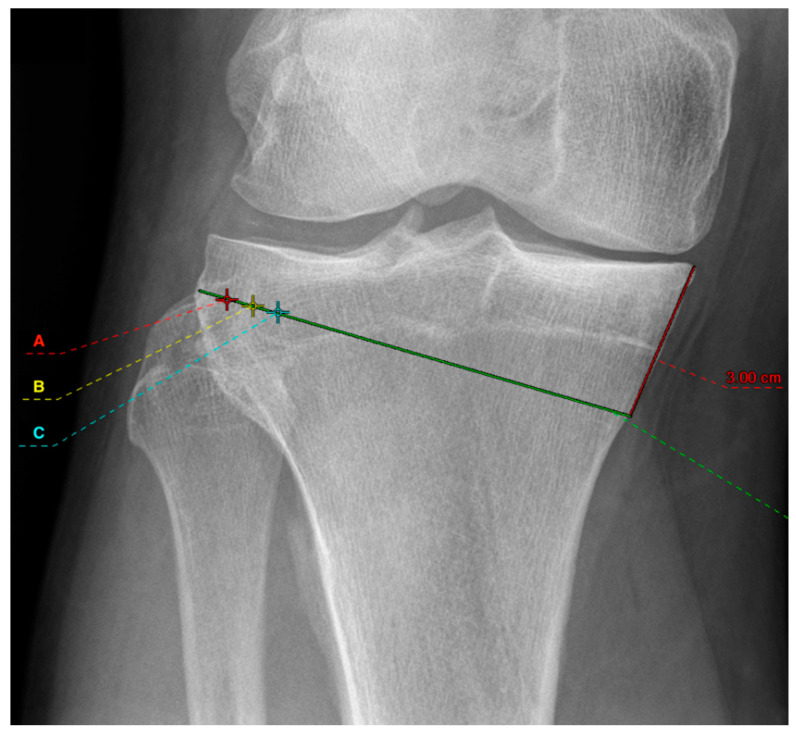
Illustration of the medial osteotomy height at 3 cm (red line) with the selected hinging points at 5 mm (red cross marked with an A), 10 mm (yellow cross marked with a B), and 15 mm (cyan cross marked with a C). The green line represents the osteotomy.

**Figure 2 diagnostics-12-02546-f002:**
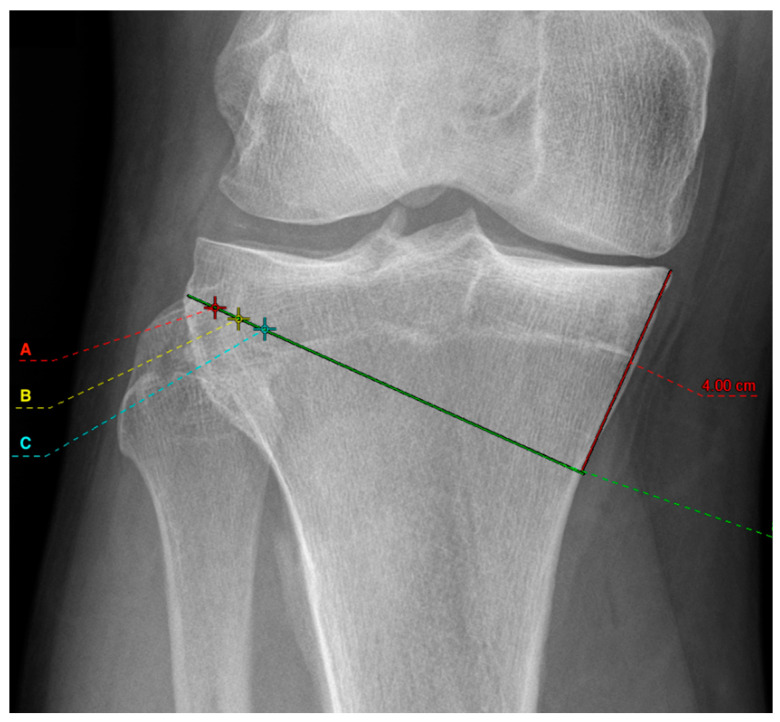
Illustration of the medial osteotomy height at 4 cm (red line) with the selected hinge points at 5 mm (red cross marked with an A), 10 mm (yellow cross marked with a B), and 15 mm (cyan cross marked with a C). The green line represents the osteotomy.

**Figure 3 diagnostics-12-02546-f003:**
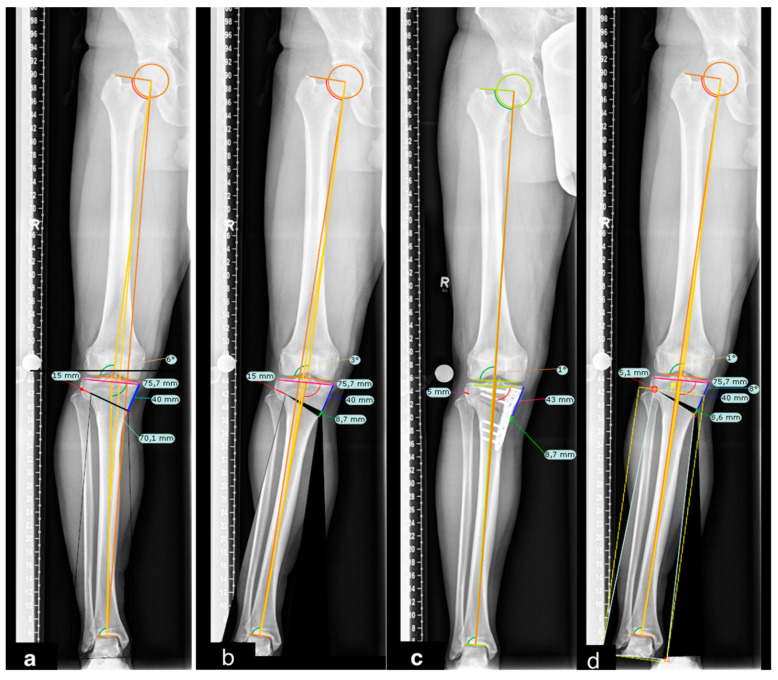
Full-weight bearing long-leg radiographs: (**a**) 6°-varus knee preoperative planning with hinge point at 15 mm from lateral cortex and sawing point at 40 mm from medial tibial plateau. (**b**) planned correction at 55% of the tibial plateau measured from medial resulting in 8.7 mm tibial wedge. (**c**) postoperative result showing an under-correction with a hinge point at 5 mm from the lateral cortex and a tibial wedge of 8.7 mm resulting in 1° of valgus. (**d**) simulation with a hinge point at 5 mm from lateral cortex and a tibial wedge of 8.6 mm resulting in 1° of valgus. (yellow lines) femoral and tibial mechanical angle. (orange line) Mikulicz line; (red line) distance from lateral cortex to hinge. (blue line) distance from the tibial plateau and sawing point. (green line) wedge width. (pink line) tibial plateau mediolateral width.

**Table 1 diagnostics-12-02546-t001:** Measurement results.

	Mean Wedge Width (mm)	SD
3 cm-A	8.9	±3.9
3 cm-B	10.3	±4.2
3 cm-C	11.6	±4.4
4 cm-A	9	±3.8
4 cm-B	10.2	±4.1
4 cm-C	11.4	±4.2
plateau width (mm)	79.5	±6.9
femoral length (mm)	475.0	±34.5
tibial length (mm)	372.6	±30.2

Distance to the lateral cortical surface A = 0.5 cm B = 1.0 cm C = 1.5 cm, 3 cm = osteotomy level 3 cm below the tibial plateau, 4 cm = osteotomy level 4 cm below the tibial plateau.

**Table 2 diagnostics-12-02546-t002:** Distribution of varus knees.

Varus (°)	Frequency (n = 55)	Percent (%)
3.00	15	27.3
4.00	10	18.2
5.00	4	7.3
6.00	8	14.5
7.00	2	3.6
8.00	5	9.1
9.00	6	10.9
12.00	3	5.5
13.00	1	1.8
17.00	1	1.8

**Table 3 diagnostics-12-02546-t003:** Intraobserver reproducibility and interobserver reliability.

Observers	Rater 1 ICC (95% CI)	Rater 2 ICC (95% CI)	R1 vs R2 ICC (95% CI)
mFTA	0.988 (0.972–0.995)	0.988 (0.972–0.995)	0.988 (0.972–0.995)
3 cm-A	0.990 (0.975–0.996)	0.990 (0.977–0.996)	0.979 (0.954–0.991)
3 cm-B	0.995 (0.990–0.998)	0.994 (0.986–0.997)	0.990 (0.976–0.995)
3 cm-C	0.996 (0.991–0.998)	0.992 (0.982–0.997)	0.988 (0.972–0.995)
4 cm-A	0.993 (0.983–0.997)	0.994 (0.968–0.998)	0.988 (0.864–0.997)
4 cm-B	0.996 (0.990–0.998)	0.995 (0.983–0.998)	0.991 (0.943–0.997)
4 cm-C	0.994 (0.984–0.997)	0.992 (0.979–0.997)	0.985 (0.942–0.995)
plateau width	0.994 (0.979–0.998)	0.955 (0.894–0.980)	0.949 (0.843–0.981)
femoral length	0.998 (0.996–0.999)	0.998 (0.993–0.999)	0.997 (0.981–0.999)
tibial length	0.996 (0.991–0.998)	0.830 (0.651–0.922)	0.827 (0.644–0.921)

Mfta = mechanical femoral tibial axis; Distance to the lateral cortical surface A = 0.5 cm B = 1.0 cm C = 1.5 cm, 3 cm = osteotomy level 3 cm below the tibial plateau, 4 cm = osteotomy level 4 cm below the tibial plateau, R1 = Rater 1, R2 = Rater 2.

**Table 4 diagnostics-12-02546-t004:** Tukey’s HSD Test for multiple comparisons of the wedge width between the groups.

Compared Groups	Difference of Mean (mm)	*p*	Rate ≥ 2 m (%)
3 cm-A vs. 3 cm-B	1.325	0.534	9%
3 cm-B vs. 3 cm-C	1.285	0.568	7%
3 cm-A vs. 3 cm-C	2.611	0.012 *	91%
4 cm-A vs. 4 cm-B	1.216	0.626	4%
4 cm-B vs. 4 cm-C	1.198	0.641	2%
4 cm-A vs. 4 cm-C	2.415	0.026 *	82%
3 cm-A vs. 4 cm-A	0.009	1.000	0%
3 cm-A vs. 4 cm-B	1.225	0.619	7%
3 cm-A vs. 4 cm-C	2.424	0.025 *	78%
3 cm-B vs. 4 cm-A	1.316	0.541	7%
3 cm-B vs. 4 cm-B	0.100	1.000	0%
3 cm-B vs. 4 cm-C	1.098	0.722	0%
3 cm-C vs. 4 cm-A	2.601	0.012 *	80%
3 cm-C vs. 4 cm-B	1.385	0.483	16%
3 cm-C vs. 4 cm-C	0.187	1.000	0%

(*) *p* < 0.05 = significant. Distance to the lateral cortical surface A = 0.5 cm B = 1.0 cm C = 1.5 cm, 3 cm = osteotomy level 3 cm below the tibial plateau, 4 cm = osteotomy level 4 cm below the tibial plateau.

## Data Availability

Not applicable.

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
