# Peer review of "Influence of Medial Osteotomy Height and Hinge Position in Open Wedge High Tibial Osteotomy: A Simulation Study"

_diagnostics, 2022, doi:10.3390/diagnostics12102546_

Round 1

Reviewer 1 Report (Previous Reviewer 1)

The topic is very interesting. The study is very well designed and the paper well written.

I would further describe more in details how parameters were considered, in particular different hinge points and their effect on clinical practice.

Discussion: please discuss more in details possible clinical effects of these findings. How could these findings help in clinical practice?

Author Response

Dear reviewer 1,

Thank you for your comments on our manuscript. We have addressed them in the manuscript. You will find adjustments and corrections to the comments below.

Moderate English changes required.

Moderate English changes have been done by a native speaker.

The topic is very interesting. The study is very well designed and the paper well written.

I would further describe more in details how parameters were considered, in particular different hinge points and their effect on clinical practice.

Thank you for your comment. The following section was added.

(Line 91 - 95): «In the literature, different hinge points are recommended [38]. The results were also different concerning the correction accuracy and the risk of hinge fractures. The study of Nakamura [38], however, could show that positioning the hinge point in a “safe zone” reduces the risk of hinge fracture. For this reason, the different hinge points in this simulation study were all selected in the "Safe Zone" area.»

Discussion: please discuss more in details possible clinical effects of these findings. How could these findings help in clinical practice?

Thank you for your comment. The following section was added.

(Line 432 - 439): “Thus, it can be hypothesized that the hinge point should be set at 5mm medial to the lateral corticalis of the tibial plateau to avoid under- or overcorrection. However, this hypothesis needs to be tested by further studies. Furthermore, there was no relevant difference regarding the medial sawing point. This therefore allows for some intraoperative flexibility. Nevertheless, the results support the need for digital preoperative planning in order to increase the success rate and reduce unintended outcomes of this procedure.”

with kind regards

Reviewer 2 Report (New Reviewer)

As medial open wedge HTO gained interest in the last years, a large number of procedures were performed. In order to increase the success rate and reduce complications of this procedure, every detail regarding the surgical technique should be refined. Therefore this study is of great interest. Successful surgery relies on both good planning and a well-performed operation. Thus your conclusion regarding the variability of the correction in relation to the hinge position is of main interest and has practical application.

Author Response

Dear reviewer 2,

Thank you for your comments on our manuscript. We have addressed them in the manuscript. 

English language and style are fine/minor spell check required

Minor spell check has been performed by a native speaker .

As medial open wedge HTO gained interest in the last years, a large number of procedures were performed. In order to increase the success rate and reduce complications of this procedure, every detail regarding the surgical technique should be refined. Therefore this study is of great interest. Successful surgery relies on both good planning and a well-performed operation. Thus your conclusion regarding the variability of the correction in relation to the hinge position is of main interest and has practical application.

Thank you very much for the positive feedback on our study.

with kind regards

This manuscript is a resubmission of an earlier submission. The following is a list of the peer review reports and author responses from that submission.

Round 1

Reviewer 1 Report

The topic is very interesting. The study is very well designed and the paper well written.

I would describe more in details how parameters were considered, in particular different hinge points and their effect on clinical practice.

figure 1 is confusing. Legend should be more informative.

Discussion: please discuss more in details possible clinical effects of these findings. How could these findings help in clinical practice?

Author Response

Dear reviewer 1

Thank you very much for your constructive criticism. We have worked on all your reviews and we are glad that we could improve the manuscript with your help. The answers to your comments can be found in the appendix.

With kind regards

Alexander Korthaus

“The topic is very interesting. The study is very well designed and the paper well written.”

Dear reviewer, thank you very much, we are pleased that you like our study.

“I would describe more in details how parameters were considered, in particular different hinge points and their effect on clinical practice.”

We adjusted in line 52-53 and 75-76 the sentences leading to better understanding why we choose the two osteotomy heights and the different hinge points. Line 192-195 explains the effect on clinical practice. 

“figure 1 is confusing. Legend should be more informative.”

Thank you for the comment. We have adjusted the legend of the figure.

“Discussion: please discuss more in details possible clinical effects of these findings. How could these findings help in clinical practice?“

Line 192-195 explains how these findings could help in clinical practice.  In general the findings show that planning can help avoiding under- or over-correction which is addressed in various paragraphs of the discussion (line 212-215, 229-230, 246-248, 254-255, 267-272)

Reviewer 2 Report

The authors conducted an interesting study on the effect of hinge position on wedge height based on radio graphic measurements. This is of clinical implication. However, the manuscript needs to be further polished. Ia have several comments as follows: 

  1. Please address why this study is important for readers in the introduction.  
  2. Where is table 1 first cited in the manuscript? Please clarify. 
  3. Please merge table 2 and 3 as one table. 
  4. Please display the difference of mean in table 4, not just P values, as P values are of no clinical importance here. 
  5. Please clarify how the length of femur and tibia was measured.
  6. The authors defined a difference of > 2 mm between the measured wedges high as clinically important difference. Please give the rate of reaching this difference between each pair compared groups and is there any significant difference between groups. 
  7. Please give the rate of achieving the planned correction of each group and is there any significant difference between groups. 
  8. Please discuss the possibility of fracture and failure to achieve proper correction of each predefined group according to literature. 
  9. This study is merely conducted based on radiographs, with no real world data. Please acknowledge the limitation of such a simulated study.
  10. Page 2, line 74 and 75, high should be height? 
  11. There are two figure 1s. Please provide the correct number and order of citation. 
  12. Please clarify why and how this study is waived for ethical approval. 

Author Response

Dear reviewer 2

Thank you very much for your helpful criticism. We have incorporated all of your comments and are pleased that we were able to improve the manuscript with your help. The answers to your comments can be found in the appendix.

With kind regards

Alexander Korthaus

“The authors conducted an interesting study on the effect of hinge position on wedge height based on radio graphic measurements. This is of clinical implication. However, the manuscript needs to be further polished. I have several comments as follows:“ 

  1. Please address why this study is important for readers in the introduction.  

Thank you, the Introduction hast been chanced (line 55-58)

  1. Where is table 1 first cited in the manuscript? Please clarify. 

It is citied in Line 115.

  1. Please merge table 2 and 3 as one table. 

We merged the two table

  1. Please display the difference of mean in table 4, not just P values, as P values are of no clinical importance here. 

We added the values in table 4

  1. Please clarify how the length of femur and tibia was measured.

Thank you we added a sentence in line 78-80

  1. The authors defined a difference of > 2 mm between the measured wedges high as clinically important difference. Please give the rate of reaching this difference between each pair compared groups and is there any significant difference between groups. 

We added the values in table 4. The significance between the groups was added as well.

  1. Please give the rate of achieving the planned correction of each group and is there any significant difference between groups. 

We are sorry but, since this study is a simulation on the basis of anonymized data, no information can be given about the achievement rate of the planned correction.

  1. Please discuss the possibility of fracture and failure to achieve proper correction of each predefined group according to literature. 

Thank you for pointing this out. We have addressed this in the discussion.

  1. This study is merely conducted based on radiographs, with no real world data. Please acknowledge the limitation of such a simulated study.

Thank you, we have adjusted the limitations.

  1. Page 2, line 74 and 75, high should be height? 

Thank you! We chanced it.

  1. There are two figure 1s. Please provide the correct number and order of citation. 

Thank you for the hind. We chanced it.

  1. Please clarify why and how this study is waived for ethical approval. 

This study was conducted retrospectively using anonymized data. The data were already collected prior to the study as part of clinical practice. Therefore, according to the regional ethics committee, consultation is not necessary. A statement to this effect has been added to the paper.

Round 2

Reviewer 2 Report

Thanks for the revisions. I read the revised manuscript with interest. The following comments are for the authors' consideration: 

1. In this simulation study using merely lower extremity full length films, only coronal alignment can be measured. Accurate measures can only be achieved with radiographs taken in a good position. For example, if the leg rotated externally or internally, the coronal alignment can be improperly measured. Therefore, please clarify how many of the included films are eligible for proper measurements, and the criteria to determine the quality of the films and eligibility for inclusion. 

2. The presentation of Table 2 can be improved. For example, the measures of inter- and intra-observer reliability can be displayed side by side. 

3. Please provide more data on the characteristics of the sample, such as the distribution of coronal alignment. 

4. Is baseline coronal alignment associated with the difference in mean and the rate of >=2mm ?